# Effects of Inflammatory Factor Expression Regulated by 12/15 Lipoxygenase on Obesity-Related Nephropathy

**DOI:** 10.3390/nu14132743

**Published:** 2022-06-30

**Authors:** Nian Liu, Yang Liu, Dan Dong, Jinyu Yu, Hang Yuan

**Affiliations:** 1Department of Urology, First Hospital of Jilin University, Xin Min Street 1, Changchun 130021, China; liu_nian@jlu.edu.cn; 2Department of Nephrology, First Hospital of Jilin University, Xin Min Street 1, Changchun 130021, China; ly840604@jlu.edu.cn (Y.L.); Sky_811@jlu.edu.cn (D.D.); yujinyu@jlu.edu.cn (J.Y.)

**Keywords:** obesity related nephropathy, 12/15 lipoxygenase, inflammatory factors, histone methylation

## Abstract

Background: It has been demonstrated that 12/15-lipoxygenase (LO) contributes to insulin resistance by promoting beta cells’ exposure to inflammation. We investigate the mechanism by which 12/15-LO regulates the expression of inflammatory factors in obesity-related glomerular disease (ORG). Methods: Glomerular mesangial cells were treated with metabolite of 12/15-LO, and the expression of inflammatory factors was measured. Cell histones methylation in 12/15-LO related metabolic memory process were evaluated by chromatin immunoprecipitation (ChIP) assays. Wild-type (WT) and 12/15-LO knockout mice were fed a high-fat diet (HFD) to induce ORG. Results: 12(S)-HETE increased TNF-α, MCP-1, and IL-6 mRNA expression. Inhibition of 12/15-LO reduced the expression of inflammatory factors stimulated by PA or TNF-α. ChIP assays showed that 12(S)-HETE increased H3K4me modification in the TNF-α, IL-6, and MCP-1 gene promoters, and decreased H3K9me3 modification in the MCP-1 and IL-6 gene promoter. Urinary albumin excretion was greater in HFD-fed than in standard fat diet-fed mice, but both urinary protein and microalbumin amounts were lower in HFD-fed 12/15-LO knockout than in WT mice. The levels of TNF-α, IL-6, and MCP-1 in serum and renal cortex were higher in WT than in 12/15-LO knockout mice. Conclusions: 12/15-LO may regulate the expression of inflammatory factors in ORG by methylation of histones in the promoter regions of genes encoding inflammatory factors, sustaining the inflammatory phenotype of ORG.

## 1. Introduction

Obesity can shorten life expectancy and endanger human health. Central obesity is a major risk factor for diabetes and hypertension, both of which are observed in 70% of patients with end-stage renal disease. Persistent consumption of a high-fat diet (HFD) and hyperlipidemia can lead to low-level chronic inflammation, as shown by infiltration of inflammatory cells into adipose tissue, kidneys, and other tissues, leading to oxidative stress, cytokine activation, and adipocytokine secretion. Excess adipocytes deposited in other tissues secrete inflammatory factors, further aggravating cell and tissue damage and leading to albuminuria, insulin resistance, diabetes, and atherosclerosis [1,2].

In addition to its function in energy storage, adipose tissue produces a variety of adipokines, such as TNF-α and adiponectin [3,4]. The balance between these adipokines is disrupted in obese individuals, generating a pro-inflammatory environment that leads to insulin resistance, obesity-related glomerulopathy (ORG), and renal hemodynamic abnormalities. These imbalances manifest as renal lesions, which are characterized by endothelial and podocyte disorders, thickening of the glomerular basement membrane with mesangial matrix hyperplasia, and interstitial fibrosis, resulting in a progressive decrease in renal function [5]. Thus, chronic inflammation and the overexpression of inflammatory factors play important roles in the course of ORG.

The enzyme 12/15-lipoxygenase (12/15-LO) inserts oxygen molecules into polyunsaturated fatty acids to generate products such as hydrogen peroxide [6]. The 12/15-LO enzyme can induce insulin resistance and is involved in the development and progression of diabetes and diabetic nephropathy by regulating inflammatory factors [7,8,9]. In non-obese diabetic mice, 12/15-LO inhibitors can reduce islet inflammation and hyperglycemia [10]. Although 12/15-LO is also associated with obesity-associated oxidative stress and islet dysfunction [11], its relationships to ORG and pathological injury mechanisms have not been determined. The present study investigated the effects of 12/15-LO gene knockout (LOKO) in mice to see the effects of 12/15-LOKO on ORG progression, pathological damage, and intrarenal expression of inflammatory factors. In addition, the effects of the 12/15-LO pathway on intracellular expression of inflammatory factors were analyzed in mesangial cells (MCs) to assess the role of 12/15-LO-induced chemical modifications of histone in regulating the expression of inflammatory factors.

## 2. Materials and Methods

### 2.1. Cell Culture

Primary MCs were isolated from the glomeruli of wild-type (WT) and 12/15-LOKO mice and cultured in RPMI-1640 medium (Gibco, New York, NY, USA), and monocytes (RAW264.7) were purchased from national cell line resource bank Shanghai Branch, cultured in DMEM medium (Gibco, New York, NY, USA) and supplemented with penicillin G (100 U/mL), streptomycin (100 mg/mL), and 10% fetal bovine serum (Gibco, New York, NY, USA). MCs were changed to serum-free medium when they fused and covered 70–80% of the plate, and then cultured for another 24 h to induce synchronous culture. In gene silencing experiments, cells were transfected with TNF-α siRNA or 12/15-LO siRNA (Integrated DNA Technologies, Coralville, LOWA, USA) separately, or Silencer Negative Control siRNA (siNTC) using a Nucleofector Kit (Lonza, Basel, Switzerland). After transfection efficiency evaluation, the cells were placed in medium (RPMI 1640 + 0.2% bovine serum albumin) for 24 h, and then treated with 10 ng/mL TNF-α (R&D Systems, Minneapolis, MN, USA) or 0.1 µM 12(S)-HETE (Cayman, Ann Arbor, MI, USA). Total RNA was extracted and mRNA levels were detected by RT-qPCR. Stable overexpression of mouse 12/15-LO in an immortalized MMC cell line (from Dr. Kumar Sharma, Thomas Jefferson University, Philadelphia, PA, USA) was performed as described recently [12]. TNF-α, MCP-1, IL-6, and IL-10 protein levels in the primary MC supernatants were measured with precoated ELISA kits (DAKEWE, 1217202, 1217392, 1210602, 1211002) according to the manufacturer’s protocol.

### 2.2. Experimental Animals, Groups, and Administration

Fourteen 8-week-old C57BL/6 WT mice and 14 age-matched 12/15-LOKO mice were purchased from Jackson Laboratories of America. All animal experiments were approved by the Ethics Committee of the First Hospital of Jilin University and conformed to the internationally accepted principles for the care and use of laboratory animals. The mice were fed a standard diet for 1 week, and six mice in each group were randomly assigned to be fed a SFD or a HFD (Readybiotech company, Shenzhen, China). Body weight and fasting blood glucose were analyzed every week, and 24 h urine samples were collected in metabolic cages at 1, 8, and 14 weeks.

### 2.3. Sample Collection

After 14 weeks, the mice were intraperitoneally anesthetized with 350 mg/kg avertin and sacrificed. Their blood and kidneys were harvested. Serum was isolated by centrifuge, and adipokine and insulin levels were assessed using ELISA kits (Abcam, Boston, MA, USA). The kidneys were weighed, and the length of each tibia was measured. Urine samples were diluted 5X with Milli Q distilled water and the amount of protein (in milligrams) was checked using detergent-compatible protein assay kits (Bio-Rad, Inc., Hercules, CA, USA) as described by the manufacturer’s instructions. We measured the ratio of urinary albumin to creatinine (ACR) with an indirect competitive ELISA kit (Exocell, Philadelphia, PA, USA) according to the manufacturer’s instructions. The collected kidney sections were fixed in 10% formalin for histological examination.

### 2.4. Glucose Tolerance Test (GTT)

Mice fasted for at least 10 h were intraperitoneally injected with 5 g/kg body weight of glucose dissolved in 0.9% NaCl. Blood samples from caudal vein of experimental mice were withdrawn before injection and 30, 60, 90, and 120 min after injection to measure blood glucose concentrations.

### 2.5. Western Blotting Analysis

Western blotting was performed as described in [13] using primary antibodies against 12/15-LO (Abcam, Boston, MA, USA); TGF-β1 (R&D Systems, Minneapolis, MN, USA); PAI-1 (BD Biosciences, San Jose, CA, USA); and β-actin (Sigma, Saint Louis, MO, USA), each diluted 1:1000, and a 1:1000 dilution of HRP-conjugated secondary antibody (Sigma, Saint Louis, MO, USA).

### 2.6. Chromatin Immunoprecipitation (ChIP Assay)

MCs were incubated with 0.1 µM 12(S)-HETE for 0.5–4 h, and then fixed with 1% formaldehyde for 10 min (min) at room temperature, followed by neutralization with 125 mM glycine for 5 min. After being washed with cold phosphate-buffered saline (PBS), fixed cells were lysed using 1% SDS lysis buffer supplemented with protease inhibitors. Cell lysates were sonicated and preincubated with protein A/G immunoprecipitate magnetic beads (Invitrogen; Thermo Fisher Scientific, Inc., San Jose, CA, USA), and then specific anti-H3Kme antibody was added in (Abcam, Boston, MA, USA) overnight at 4 °C. The precipitated DNA fragments were prepared as templates for quantitative PCR on an ABI-7500 real-time quantitative thermal cycler. Results were quantitated using the 2^−ΔΔCt^ method.

### 2.7. Reverse Transcription-Quantitative Polymerase Chain Reaction (RT-qPCR) Analysis

Total RNA was extracted from cells and tissue samples using RNA STAT60 kits (Tel-Test, Friendswood, TX, USA). After reverse transcription, real-time PCR was performed using SYBR Green reagent (Applied Biosystems, Foster City, CA, USA). Each amplification reaction included 10 µL SYBR Green mix, with 1 µL containing 50 pmol of each primer (Table 1), 5 µL cDNA template, and 3 µL DEPC water, to a total volume of 20 µL. The level of each mRNA was normalized to that of β-actin mRNA using the 2^−ΔΔCt^ method.

### 2.8. Histologic Staining and Morphometric Analyses

Renal cortex sections were fixed in 10% buffered formalin solution for 24 h, dehydration by gradient alcohol, embedded in paraffins, and then cut into 5 um sections for periodic acid-Schiff (PAS) and immunohistochemistry (IHC).

The slices were oxidized with 1% periodic acid for 10 min, and then soaked in Schiff reagent for 40 min, followed by stain in hematoxylin staining solution for 2 min. Images were captured using an Olympus BX53 Microscope Digital Camera and were processed with CellSense Entry software. Twenty glomeruli were randomly selected to evaluate glomerular hypertrophy and extracellular matrix (ECM) deposition. The extent of mesangial matrix identified by PAS-positive material, glomerular area, and mesangial matrix area was measured with image-analysis software (Image-Pro Plus 5.1, Media Cybernetics, Silver Spring, MD, USA) and analyzed as described previously [13]. IHC was performed for macrophage staining. Briefly, slices were incubated for 30 min at room temperature with a monoclonal rabbit anti-mouse F4/80 antigen antibody (cell signaling, 70076) diluted 1:200 with 2% casein in BSA. After washing, secondary goat anti-rabbit antibody was added for 15 min. Positive stained areas in glomeruli were measured using Image-Pro Plus 5.1.

### 2.9. Statistical Analyses

Data were expressed as mean ± standard error mean (SEM) from multiple experiments. Paired Student’s t-tests were used to compare two groups or ANOVA with Dunnett’s post-test for multiple groups using PRISM5 software (Graph Pad, San Diego, CA, USA). *p* < 0.05 was considered statistically significant.

## 3. Results

### 3.1. Effects of 12/15-LO and Its Metabolite 12(S)-HETE on the Expression of Adipose Inflammatory Factors

MCs derived from mouse glomeruli were stimulated with the 12/15-LO metabolite 12(S)-HETE (0.1 µM) for 1–8 h and intracellular mRNA expression of TNF-α, IL-6, MCP-1, and IL-10 were measured. 12(S)-HETE stimulation was found to significantly increase intracellular TNF-α mRNA levels, which peaked at 4 h (*p* < 0.001) (Figure 1A), as well as significantly increasing the mRNA levels of IL-6 and MCP-1, which are regulated by TNF-α (*p* < 0.01) (Figure 1B,C). 12(S)-HETE also slightly reduced IL-10 mRNA levels, but the difference was not statistically significant (Figure 1D). Next, we assessed the effect of 12(S)-HETE on secretion of adipose inflammatory factors into the medium with ELISA. 12(S)-HETE significantly increased TNF-α, IL-6, and MCP-1 concentration released by MC (Appendix A). Other pro-inflammatory gene (IL-12, IL-1β, CXCL10) mRNA expressions were also detected; there was no obvious difference in response to 12(S)-HETE (Appendix A). To further assess the relationship between 12/15-LO and the expression of inflammatory factors, we stably overexpressed of 12/15-LO in an immortalized MMC cell line as described previously [12]. The levels of 12/15-LO protein were higher in cells transfected with 12/15-LO DNA than with control pcDNA vector (Figure 1E), as were the levels of TNF-α, IL-6, and MCP-1 mRNAs (Figure 1F) and their secretion in the supernatant of MC (Figure 1G). 

### 3.2. Effect of siRNA Inhibition of 12/15-LO Expression on Palmitic Acid-Regulated Inflammatory Factor Expression

Treatment of MCs and monocyte (MO) cell line (RAW264.7) with 10 mM palmitic acid (PA), which simulates high-fat stimulation, significantly increased 12/15-LO mRNA levels in both cell types (*p* < 0.001) (Figure 2A). Transfection of these cells with an siRNA specific for the 12/15-LO gene significantly reduced intracellular 12/15-LO mRNA and protein expression, as shown by RT-qPCR and Western blotting assays, respectively (Figure 2B,C). Gene silence of 12/15-LO by siRNA did not change pro-inflammatory factors level compared with siNTC; treatment of these cells with 10 mM PA resulted in lower levels of mRNAs encoding the pro-inflammatory factors TNF-α, IL-6, and MCP-1 than in PA-treated cells transfected with siNTC (Figure 2D), suggesting that PA-induced expression of inflammatory factors may be mediated by 12/15-LO.

### 3.3. Effect of Inhibition of 12/15-LO Expression on TNF-α-Regulated Inflammatory Factor Expression

TNF-α, which was the first identified co-effector of inflammation and obesity, is involved in PA-induced regulation of inflammatory factors and insulin resistance [14]. To determine whether 12/15-LO is involved in TNF-α-regulated inflammatory factor expression, MCs transfected with an siRNA specific to 12/15-LO were treated with TNF-α (10 ng/mL) for 2 h. Inhibition of 12/15-LO partially reduced the effects of TNF-α on the expression of target genes encoding MCP-1 and IL-6, instead of IL-10 (Figure 3A), suggesting that TNF-α may contribute to the regulation of inflammatory factors, at least in part, through the 12/15-LO pathway. To further confirm these effects of 12/15-LO on TNF-α, MCs derived from WT and 12/15-LOKO mice were incubated with TNF-α for 2 h. Although TNF-α increased the levels of MCP-1 and IL-6 mRNAs in MCs from WT mice, LO gene knockout reduced the expression of these genes induced by TNF-α (Figure 3B). We also evaluated the effect of TNF-α on secretion of inflammatory factors to the supernatant with ELISA. Incubation with TNF-α significantly increased MCP-1 and IL-6 production in WT MC, though this induction was attenuated in LOKO MC (Figure 3C).

### 3.4. Effect of TNF-α Inhibition on 12/15-LO-Regulated Inflammatory Factors

Although the 12/15-LO metabolite 12(S)-HETE has been found to enhance the expression of TNF-α in MCs and MOs (RAW264.7), which directly stimulates the expression of inflammatory factors of IL-6 and MCP-1 (Figure 4), it was not clear whether 12/15-LO activates the inflammatory pathway by directly increasing TNF-α expression. The effect of 12(S)-HETE on intracellular inflammatory factors was assessed by inhibiting TNF-α expression in MOs through transfection of a specific siRNA. Transfection of TNF-α siRNA not only reduced intracellular TNF-α expression (Figure 4A), but also significantly reduced the levels of MCP-1, and IL-6 mRNAs when compared with MOs transfected with NTC-siRNA (Figure 4B). Transfection of TNF-α siRNA also inhibited the ability of 12(S)-HETE to promote inflammatory factor expression. 

### 3.5. Histone Methylation Participates in the Regulation of Inflammatory Factor Expression by 12/15-LO Metabolites

To determine whether histone methylation (Me) is involved in the ability of 12(S)-HETE to regulate the expression of inflammatory factors, chemical modifications of histones binding to the promoter and body of target genes were measured by ChIP assays. Precipitated DNA fragment was amplified by qPCR using primers (Table 1) spanning the promoter and transcription regions shown in Figure 5A. Treatment with 12(S)-HETE decreased H3K9Me3 levels at the P1 promoter regions of the MCP-1 and IL-6 genes, but not the P1 promoter regions of the CypA and TNF-α genes (Figure 5B). Because H3K9Me3 modification is closely associated with the repression of gene transcription, whereas H3K4Me modification promotes gene expression, the effects of 12(S)-HETE on H3K4Me levels were evaluated. Treatment with 12(S)-HETE increased H3K4Me1 levels at the P1 region of the MCP-1 gene and at the E1 region near the transcription start site (TSS) of the MCP-1 gene promoter (Figure 5C). Treatment with 12(S)-HETE also increased H3K4Me1 in the P1 region of the TNF-α gene promoter, while it had no effects on H3K4Me1 levels in the P1 regions of the IL-6 and CypA promoters. Evaluation of the effects of 12(S)-HETE on H3K4Me3 modifications showed that 12(S)-HETE increased H3K4Me3 in the P1 regions of TNF-α, MCP-1, and IL-6, but had no effect on the P1 region of CypA or the E1 transcribed region of MCP-1 (Figure 5D). These results demonstrate that 12(S)-HETE is involved in regulating TNF-α, MCP-1, and IL-6 gene expression by modulating histones H3K4Me and H3K9Me. 

### 3.6. Effects of 12/15-LO on Mouse Body Mass, Urinary Protein Content, and Other Indicators in Obese Mice

As shown in Table 2, mice fed a HFD for 14 weeks had a significantly higher mean body mass than mice fed a SFD group for 14 weeks (*p* < 0.001), but there were no significant differences in body mass between WT and 12/15-LOKO mice. Increased kidney volume and glomerular hypertrophy are pathological features of ORG; kidney mass was significantly higher in HFD than in SFD mice (*p* < 0.001), and the kidney/body mass index was lower in both WT and 12/15-LOKO mice fed a HFD than those fed a SFD (*p* < 0.05). Because the experimental results were affected by the excessive increase in body mass in the HFD group, the renal kidney hypertrophy index (RKHI), defined as the ratio of kidney mass to tibial length, was measured. The RKHI was significantly higher in the HFD than in the SFD group (*p* < 0.001), but did not differ significantly in WT and 12/15-LOKO mice.

Twenty-four-hour urine samples were collected from mice in metabolic cages after feeding for 1, 8, and 14 weeks, and total urine protein and urine microalbumin/creatinine ration (ACR) were measured. The 24 h urine volumes at 8 and 14 weeks were significantly lower in the HFD than in the SFD group (*p* < 0.01), whereas the differences between WT and 12/15-LOKO mice were not statistically significant. Urinary ACR and proteinuria was significantly higher after 8 weeks in WT mice fed the SFD than the HFD (*p* < 0.05). Although ACR was somewhat higher after 8 weeks in 12/15-LOKO mice fed the HFD than the SFD, this difference was not statistically significant. At 14 weeks, urinary ACR levels were significantly higher in both WT and 12/15-LOKO mice fed the HFD than the SFD (*p* < 0.05), although the differences between the WT and 12/15-LOKO mice were not statistically significant. These results suggest that obesity in HFD-induced mice can cause renal injury that presents as proteinuria and albuminuria, with both occurring later in 12/15-LOKO than in WT mice.

GTT showed that fasting blood glucose concentrations at 12 weeks were higher in mice fed the HFD than the SFD (Figure 6). After intraperitoneal injection of 5 g-kg-1 glucose for 15 min, blood glucose increased rapidly in WT mice fed the HFD, peaking at 30 min and decreasing slowly to 13.9 mmol· L-1(mM) at 120 min. Blood glucose in WT mice fed the SFD peaked 30 min after glucose injection and then decreased rapidly to approximately 8.5 mM at 90 min. In 12/15-LOKO mice fed the HFD, blood glucose peaked at 15.5 ± 2.6 mM 30 min after glucose injection and then decreased rapidly to 9.9 mM at 90 min. Fasting serum insulin concentration at 14 weeks was significantly higher in WT mice fed the HFD than the SFD (*p* < 0.05). By contrast, although the insulin concentration was higher at 14 weeks in 12/15-LOKO mice fed the HFD than the SFD, this difference was not statistically significant (Figure 7A). These results suggest that the HFD led to hyperinsulinemia and insulin resistance in WT, but not in 12/15-LOKO, mice.

### 3.7. Expression of Lipid Inflammatory Cytokines and Adiponectin in Obese Mice

After 14 weeks, serum concentrations of TNF-α and IL-6 were significantly higher (*p* < 0.01, *p* < 0.001), whereas adiponectin levels were significantly lower (*p* < 0.05) in WT mice fed the HFD than the SFD (Figure 7A). Although serum concentrations of TNF-α were also significantly higher in 12/15-LOKO mice fed the HFD than the SFD (*p* < 0.05), there were no significant differences in IL-6 and adiponectin concentrations. Serum IL-10 concentrations in both WT and 12/15-LOKO mice were not affected by the HFD. In addition, HFD altered the levels of mRNAs encoding inflammatory factors in renal tissue. Specifically, the levels of TNF-α, IL-6, and MCP-1 mRNAs were significantly higher in the renal cortices of WT mice fed the HFD than the SFD (*p* < 0.01 or *p* < 0.05), whereas 12/15-LOKO mice suppressed HFD-induced TNF-α, IL-6, and MCP-1 mRNA expression in renal tissues (Figure 7B–E). Adiponectin mRNA levels were obviously lower in the kidneys of WT mice fed the HFD than the SFD (*p* < 0.05), but did not differ significantly in 12/15-LOKO mice fed the HFD and SFD.

### 3.8. Pathological Renal Changes and Expression of Pathogenic Genes in Renal Tissues of Mice under Different Feeding Regimens

Pathologic examination of kidney samples at 14 weeks showed that the glomerular areas were greater and the mesangial areas wider, accompanied by stromal hyperplasia (PAS staining), in WT mice fed the HFD than the SFD (Figure 8A). The glomerular areas were slightly smaller in 12/15-LOKO than in WT mice fed the HFD, although this difference was not statistically significant (Figure 8B). Mesangial matrix expansion was greater in WT + HFD compared with WT + SFD (Figure 8C, *p* < 0.01); this change was not significant in LOKO mice. Next, we evaluated macrophage/monocyte infiltration in the kidney by assessing F4/80-positive cells (marker for macrophages). There were almost no staining in sections from SFD WT and LOKO mice; HFD significantly induced macrophage infiltration in the glomeruli of WT mice instead of LOKO mice (Figure 8A,D). The mRNA levels of expression of TGF-β1 and PAI-1, which are involved in extracellular matrix regulation, were significantly higher in the renal cortices of WT mice fed the HFD than the SFD (*p* < 0.05 or *p* < 0.01), but did not differ significantly in the renal cortices of 12/15-LOKO mice fed the HFD and SFD. Additionally, PAI-1 levels were significantly lower in the renal cortices of 12/15-LOKO than of WT mice (Figure 8E). Similarly, the protein levels of TGF-β1 and PAI-1 were significantly higher in the renal cortices of WT mice fed the HFD than the SFD (*p* < 0.001), but did not differ significantly in 12/15-LOKO mice fed the HFD and SFD (Figure 8F–G).

## 4. Discussion

Abnormalities in glucolipid metabolism are widespread in obese individuals and patients with diabetes and accompany the progression of various related chronic diseases. The expression of 12/15-LO was found to be significantly higher in the kidneys of obese rats and obese db/db diabetic mice [15], suggesting that 12/15-LO may be closely related to ORG pathogenesis. Inhibition of 12/15-LO expression and related metabolic pathways in diabetic mice significantly improved inflammatory factor expression and inflammatory cell infiltration into renal tissues and reduced the severity of proteinuria [9,16], confirming that 12/15-LO plays a role in inflammatory factor expression and inflammatory cell infiltration in diabetes-associated renal tissues. Related mechanistic studies have also shown that exposing primary islet cells to inflammatory factors (TNF-α, IL-1β, and IFN-γ) can lead to impaired β-cell function and survival, and that treatment with 12-HETE, a metabolite of 12/15-LO, has effects similar to those of inflammatory factors [17]. Furthermore, treatment of non-obese diabetic mice with the 12/15-LO inhibitor ML351 was found to preserve β-cell mass and function and ameliorate islet inflammation and hyperglycemia [10]. These findings suggest that 12/15-LO can affect the secretory metabolic function or apoptosis of β-cells.

The present study found that fasting blood glucose and insulin levels at 14 weeks were significantly higher in WT mice with HFD-induced obesity than in SFD-fed WT mice. By contrast, fasting blood glucose and insulin levels did not differ significantly in 12/15-LOKO mice fed the HFD and SFD, suggesting that 12/15-LO is involved in the regulation of obesity-associated insulin levels and the development of insulin resistance. GTTs also showed that abnormal glucose tolerance was improved in HFD-fed 12/15-LOKO mice compared with HFD-fed WT mice, improvements partially associated with reduced levels of inflammatory factors in adipose tissue and/or improved islet function in 12/15-LOKO mice.

The increase in adipose tissue during the development of obesity leads to abnormal production of pro-inflammatory adipokines, with TNF-α shown to be the most critical factor in the progression of adipose tissue inflammation and insulin resistance [18]. Obesity-induced insulin resistance was found to be improved in TNF-α knockout mice, and elevated levels of TNF-α were observed to enhance the production of the pro-inflammatory factors MCP-1 and IL-6, as well as downregulating the expression of anti-inflammatory adipokines such as adiponectin [19,20]. Abnormal lipid accumulation not only leads to ORG but is also closely associated with the development of chronic kidney disease (CKD) [21]. The relatively high levels of TNF-α and MCP-1, accompanied by excessive macrophage infiltration within the adipose tissue of CKD patients [22], confirm the importance of inflammatory adipokines, represented by TNF-α, in promoting the development of CKD. The present study found that, along with the significantly elevated serum TNF-α and IL-6 levels in obese mice, the levels of TNF-α, MCP-1, and IL-6 mRNAs in renal tissue were significantly higher in HFD-fed than in SFD-fed WT mice. From pathological IHC result, we observed excessive macrophage infiltration in glomerular from HFD-fed WT mice instead of HFD-fed LOKO mice. Although the levels of mRNAs encoding the pro-inflammatory factors TNF-α and MCP-1 were somewhat higher in the renal tissues of HFD-fed mice than of SFD-fed 12/15-LOKO mice, the differences were not statistically significantly, indicating that 12/15-LO knockout affects the overexpression of obesity-associated inflammatory factors such as TNF-α in the kidneys.

TNF-α, the first identified co-effector of inflammation and obesity, is significantly more elevated in adipose and muscle tissues of obese rats and humans than in their respective non-obese controls. Administration of TNF-α to experimental animals induced insulin resistance [18,21]. In addition, knockout of TNF-α or its receptor significantly enhanced insulin sensitivity and pharmacological inhibition of TNF-α attenuated obesity-induced insulin resistance [18,21]. Taken together, these results confirm the relationships among obesity, inflammation, and insulin resistance, as well as the key role of TNF-α in the development of obesity and insulin resistance. The present study found that 12/15-LO increased the levels of TNF-α, IL-6, and MCP-1 mRNAs in MCs via the metabolite 12(S)-HETE; alerting 12/15-LO expression levels via transfection significantly reduced PA associated levels of mRNAs encoding intracellular inflammatory factors such as TNF-α. In addition, lack of 12/15-LO gene decreased the induction of TNF-α in IL-6 and MCP-1 in MCs, and TNF-α partially mediated the regulatory effect of 12/15-LO on MCP-1 and IL-6 expression. From these results, we further confirmed that 12/15-LO regulates the expression of inflammatory factors: this process is highly associated with cooperation with the TNF-α pathway and these two might influence each other. Since macrophage/monocyte infiltration induces the release of proinflammaory cytokines in kidney [23], which is responsible for renal damage in ORG, it is therefore possible to mediate and amplify the 12/15-LO pathological renal injury (Figure 9). It has been reported that 12/15-LO participates in insulin resistance and macrophage infiltration in pancreatic tissue [24]; our recent study indicated that 12/15-LO engaged in the macrophage infiltration in renal tissue by diabetes [25]. In this study, we detected that 12/15-LO knockout reduced HFD-induced macrophage infiltration in glomeruli, but how to affect macrophage infiltration and the signaling pathway in MC between the TNF-α and 12/15-LO metabolic pathways is still unclear.

Despite some obese individuals and patients with diabetes having normal blood glucose and lipid levels, several pathological changes, such as microinflammation and microvascular complications, continue to progress, a phenomenon known as metabolic memory [26]. Aberrant chemical modifications of histones, which modulate the expression of certain genes, were shown to be involved in the development of metabolic memory [27,28]. Histone tails are rich in amino acid residues that can be covalently modified by, for example, acetylation (Ac), phosphorylation (Ph), and methylation (Me). These modifications alter chromatin structure, a process called chromatin remodeling, thereby affecting gene expression. Post-translational modifications of histones, particularly HKAc and HKMe, were shown to play important roles in gene expression and sustained chromatin remodeling in HFD-fed animals and those with acute kidney injury [28,29]. Animal models of ischemia-reperfusion injury were shown to be characterized by significant in vivo upregulation of the histone modification system, abundant modifications of histones binding to the promoter regions of genes encoding inflammatory factors, and significant increases in the expression of fibrotic genes and collagen [30]. 12/15-LO was shown to participate in the regulation of pathogenic gene expression in diabetic nephropathy [25] by affecting post-translational modifications of histones [13], including by increasing H3K4me1 modifications in the promoter region of the PAI-1 gene, thus promoting its expression. Gene silence of 12/15-LO by cholesterol tagged siRNA prevented macrophage infiltration in glomeruli in vivo under diabetic condition [25]. The present study found that the 12/15-LO metabolite 12(S)HETE increased the levels of mRNAs encoding intracellular inflammatory factors, such as TNF-α, IL-6, and MCP-1, and altered histone methylation in the promoter region of these genes, including increasing the levels of H3K4me1 and H3k4me3 and reducing the levels of H3K9me3 in the P1 regions of the target gene promoters. These findings suggest that 12/15-LO promotes inflammatory factor expression in part through posttranscriptional modifications of histones such as H3KMe in gene promoter regions. From the published mechanisms of epigenetic modulations under obesity or diabetic condition [13,28,31], and involved in target genes regulation including inflammatory factors [32,33,34], we noted that epigenetic H3K4me and H3K9me modifications were highly associated with metabolic obesity as well as diabetic complications; therefore, we speculate that Set7 (H3K4 methyltransferase), Suv39, or LSD1 (H3K9 methyltransferase) might be key regulatory factors in 12/15-LO triggered histone methylation around inflammatory gene promoters. Taken together, our results further confirm that by modifying histones, 12/15-LO is involved in the development of obesity-associated metabolic memory.

## 5. Conclusions

In summary, these findings confirm that 12/15-LO alters the cellular expression and secretion of inflammatory factors such as TNF-α, MCP-1, and IL-6 through its metabolites. This effect is partially mediated by increasing TNF-α levels, which, in turn, enhances the expression of MCP-1 and IL-6. Moreover, histone H3KMe is involved in the mechanism of action of 12/15-LO in regulating inflammatory factor expression.

## Figures and Tables

**Figure 1 nutrients-14-02743-f001:**
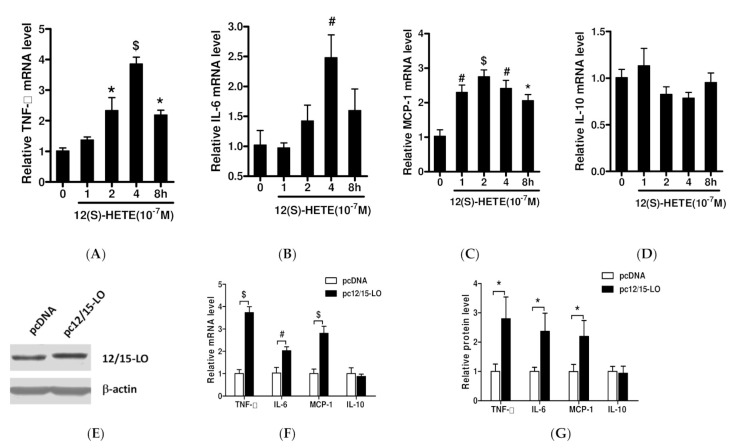
Effects of 12(S)-HETE on the expression of adipose inflammatory factors. MCs were stimulated with 12(S)-HETE (10^−7^M) for 1–8 h and changes in intracellular mRNA levels of TNF-α (**A**), IL-6 (**B**), MCP-1 (**C**), and IL-10 (**D**) were measured by RT-qPCR. Results are shown as fold over non treatment control (mean ± SEM, *n* = 3. * *p* < 0.05, # *p* < 0.01, $ *p* < 0.001 vs. control). (**E**) MCs were stably overexpressing 12/15-LO protein by transfection of 12/15-LO DNA or pcDNA as control. Equal numbers of sub-confluent pcDNA and pc12/15-LO MCs were cultured in medium (RPMI1640 + 1%FBS) for 48 h. (**F**) Intracellular mRNA expression of TNF-α, IL-6, MCP-1, and IL-10 were detected by RT-qPCR. (**G**) Aliquot supernatant was collected from each Petri dish and content of TNF-α, IL-6, MCP-1, and IL-10 were measured by ELISA kits. Results are shown as fold over pcDNA (mean ± SEM, *n* = 3. * *p* < 0.05, # *p* < 0.01, $ *p* < 0.001 vs. pcDNA).

**Figure 2 nutrients-14-02743-f002:**
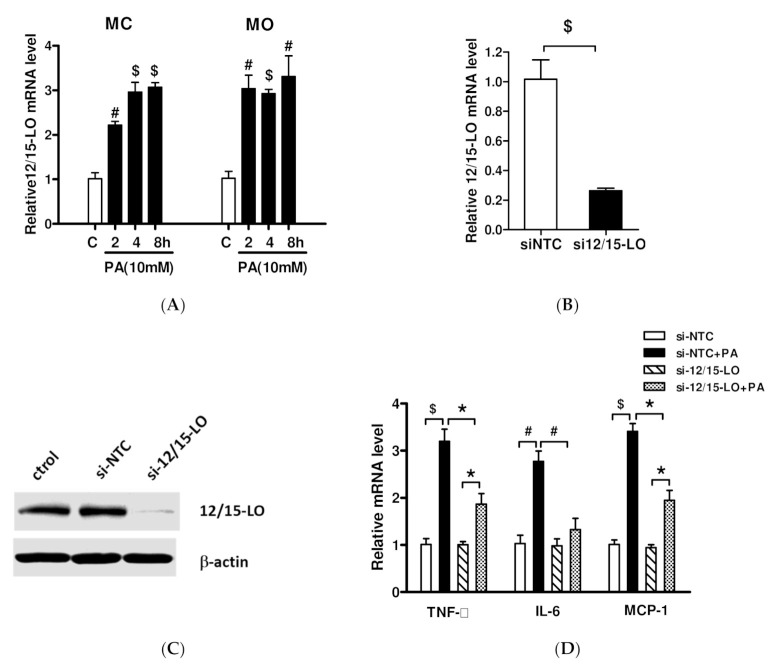
Inhibition of 12/15-LO expression on palmitic acid-regulated inflammatory factor expression. (**A**) Palmitic acid (PA, 10 mM) added to MCs and MOs to mimic high-fat stimulation and then observed intracellular 12/15-LO expression; (**B**,**C**) MCs were transfected with siRNAs specific to 12/15-LO (si-12/15-LO) or non target control (si-NTC) oligonucleotides, gene expressions were detected by WB and RT-qPCR assays. Data represented as mean ± SEM, *n* = 3. $ *p* < 0.001 versus siNTC. (**D**) Transfected MCs were added to PA (10 mM) for 8 h and gene expression of TNF-α, IL-6, and MCP-1 was analyzed by RT-qPCR. Results are expressed as fold over siNTC (mean ± SEM, *n* = 3. * *p* < 0.05, # *p* < 0.01, $ *p* < 0.001 vs.siNTC or si12/15-LO).

**Figure 3 nutrients-14-02743-f003:**
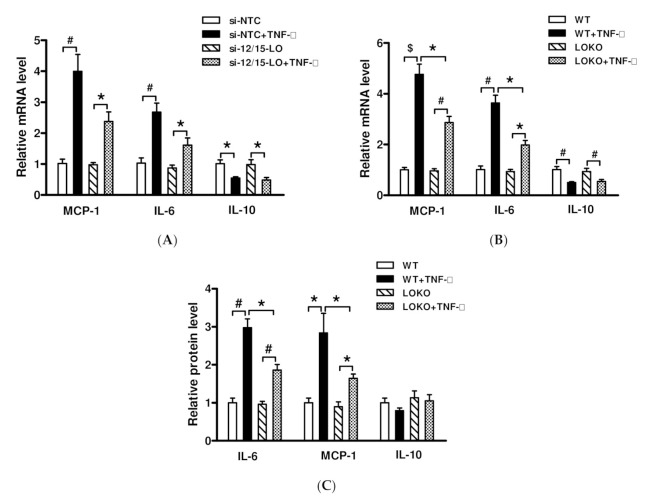
Specific 12/15-LO siRNA transfection or gene knockout attenuated TNF-α induced inflammatory genes expression. (**A**) MCs were transfected with siRNA targeting 12/15-LO or siNTC oligonucleotides; after 48 h, transfected MCs were serum depletion (SD) for 24 h, and then treated with TNF-α (10 ng/mL) for another 2 h. Expression of inflammatory genes was detected by RT-qPCR. The results are shown as fold over siNTC (mean ± SEM, *n* = 3. * *p* < 0.05, # *p* < 0.01 vs. siNTC or si12/15-LO). (**B**) MCs from WT and LOKO mice were SD for 24 h, and then treated with TNF-α (10 ng/mL) for 2 h. Expression of MCP-1, IL-6 and IL-10 was detected by RT-qPCR. (**C**) Equal numbers of sub-confluent MCs from WT and LOKO mice were cultured in the medium (RPMI1640 + 1%FBS) containing either buffer (control) or TNF-α (10 ng/mL) for 24 h. An aliquot supernatant was collected from each well and the contents of IL-6, MCP-1, and IL-10 were measured by ELISA kits. The results are expressed as fold over WT control (mean ± SEM, *n* = 3. * *p* < 0.05, # *p* < 0.01, $ *p* < 0.001 vs. control or WT + TNF-α).

**Figure 4 nutrients-14-02743-f004:**
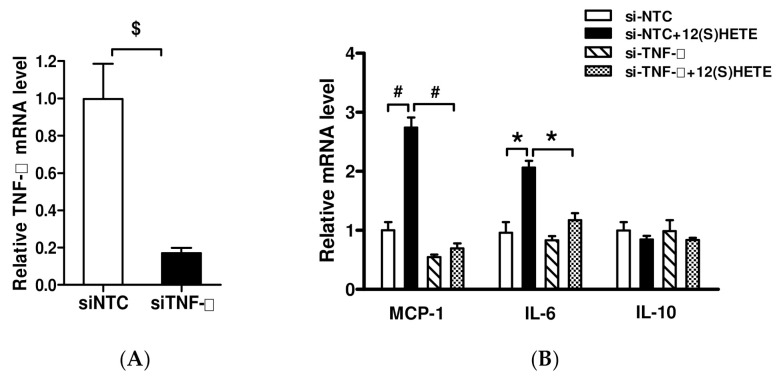
Inhibition of TNF-α expression on 12(S)HETE-activated inflammatory factor expression. (**A**) MOs were transfected with siRNAs targeting TNF-α (si-TNF-α) or non target control (si-NTC) oligonucleotides, and gene expressions were detected by RT-qPCR assays. Data represent mean ± SEM, *n* = 3. $ *p* < 0.001 versus siNTC. (**B**) Transfected MOs were SD for 24 h, then treated with 12(S)HETE for 8 h and expression of indicated genes was determined by RT-qPCR. Results are shown as fold over siNTC (mean ± SEM, *n* = 3. * *p* < 0.05, # *p* < 0.01 vs. siNTC or siTNF-α).

**Figure 5 nutrients-14-02743-f005:**
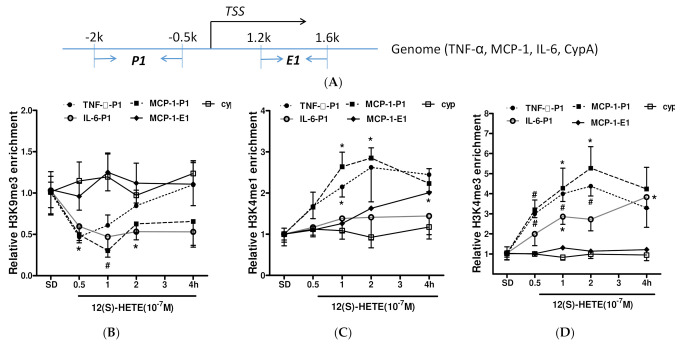
Treatment with 12(S)HETE induced histone modifications at inflammatory gene promoters. (**A**) Primers design for ChIP assay located at gene promoter (P1) or enhancer (E1) region close to transcription start site (TSS). Quiescent MMCs were treated with 12(S)HETE (10^−7^ M) for 2 h, followed by ChIP assays as described in Methods, and precipitated DNA fragments were amplified with primers targeting the inflammatory gene promoters (P) and transcription regions of enhancer (E). (**B**) 12(S)HETE (10^−7^ M) decreased occupancy of H3K9Me3 at MCP-1 and IL-6 promoter (P1); (**C**) 12(S)HETE treatment induced H3K4me1 enrichment at TNF-α promoter (P1), MCP-1 promoter (P1) as well as transcriptional enhancer (E1); (**D**) 12(S)HETE (10^−7^ M) enhanced H3K4me3 enrichment at TNF-α, MCP-1, and IL-6 promoter (P1). Results equalized to input DNA are shown as fold over untreated WT MC (mean ± SEM, *n* = 3; * *p* < 0.05, # *p* < 0.01).

**Figure 6 nutrients-14-02743-f006:**
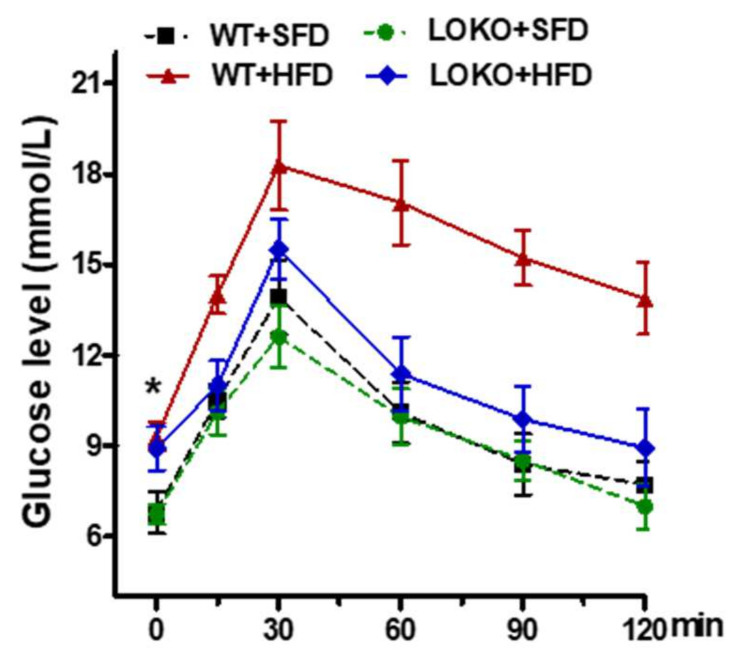
HFD WT instead of LOKO mice have reduced glucose tolerance. Blood glucose levels were tested at 30 min intervals following an intro-peritoneal glucose injection. WT HFD mice showed a higher fasting glucose level before glucose injection (* *p* < 0.05), along with a slower rate of glucose clearance compared with other groups, including LOKO HFD mice.

**Figure 7 nutrients-14-02743-f007:**
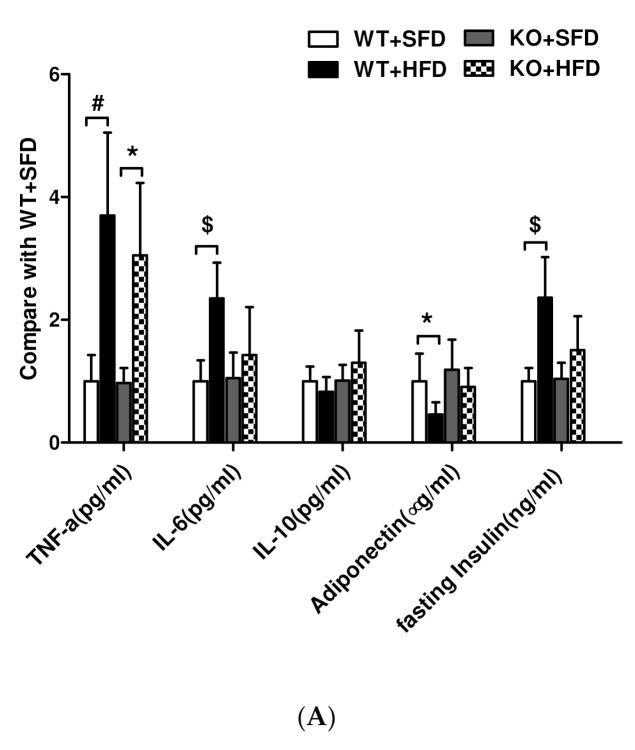
Increased inflammatory factors in obese mice were attenuated in LOKO mice. (**A**) Serum adipokine and insulin level was measured according to manufacturer’s instructions. WT HFD mice showed reduced adiponectin, increased TNF-α, IL-6, and insulin in comparison with other groups. Data represented as fold over WT control (mean ± SEM, * *p* < 0.05, # *p* < 0.01, $ *p* < 0.001 vs. WT + SFD or KO + SFD). (**B**–**E**) Total RNA isolated from kidney sections was used to analyze expression of inflammatory genes by RT-qPCR from obesity (HFD) and control (SFD) WT and LOKO mice, and is expressed as fold over WT + SFD control. Values are shown as mean ± SEM (*n* = 3). * *p* < 0.05, # *p* < 0.01.

**Figure 8 nutrients-14-02743-f008:**
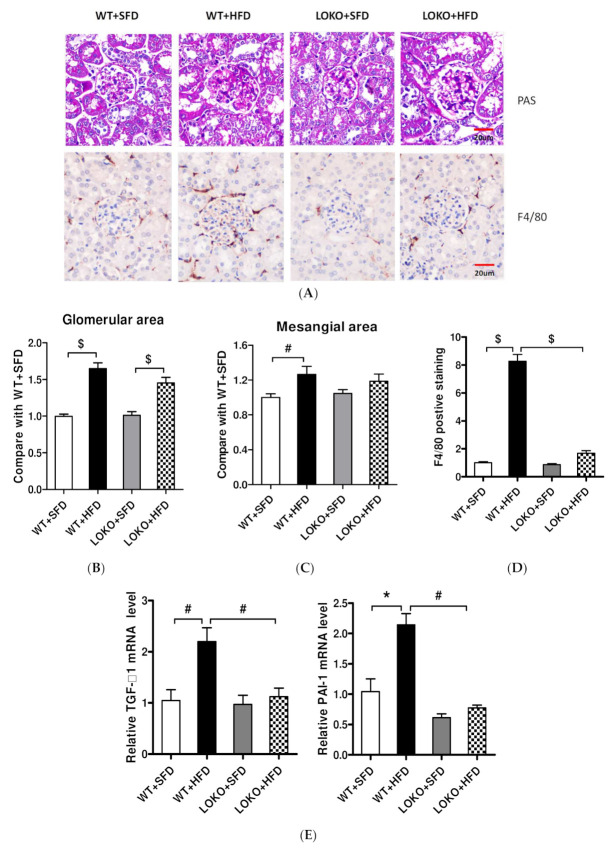
Obesity related glomerular hypertrophy, mesangial matrix expansion, and macrophage infiltration in glomeruli were less detected in 12/15-LO knockout obesity mice. (**A**) Representative images of PAS and F4/80 staining in experimental mouse glomeruli. Measurement of glomerular area (**B**), extra cellular matrix (ECM) accumulation (**C**), and F4/80-positive macrophage (**D**) in renal cortex from obesity (HFD) and control (SFD) WT and LOKO mice. Data represented as mean ± SEM. # *p* < 0.01, $ *p* < 0.001 vs. WT + SFD or LOKO + SFD. (**E**) Profibrotic genes of PAI-1 and TGF-β1 mRNA expression in renal cortical tissues were detected by RT-qPCR. (**F**) Total protein from kidneys was extracted and immunoblotted with indicated antibodies. (**G**) Band density of WB was detected and is represented as fold over WT + SFD control (mean ± SEM, *n* = 3. # *p* < 0.01, $ *p* < 0.001).

**Figure 9 nutrients-14-02743-f009:**
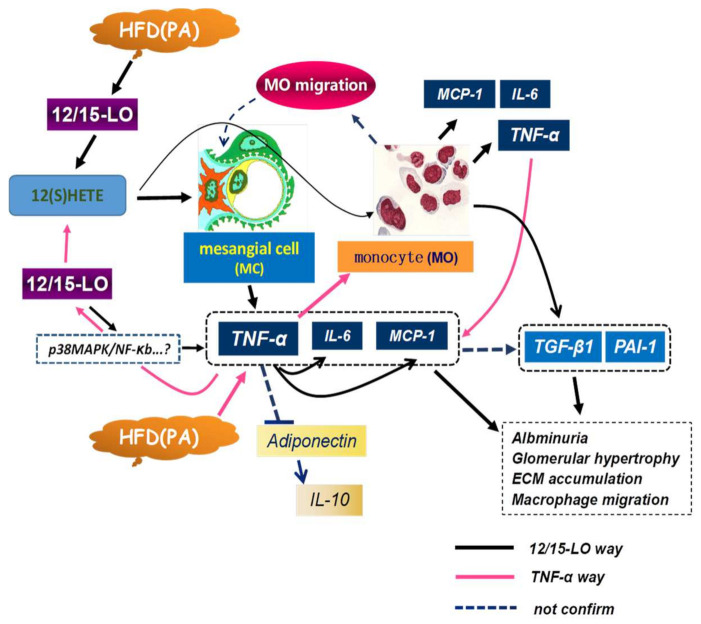
12/15-LO metabolic way in HFD induced ORG and inflammatory gene expression. 12/15-LO and its lipid product 12(S)-HETE induced mesangial cellular inflammatory gene overexpression and production. 12/15-LO mediated HFD (PA) induced inflammatory factors (TNF-α, IL-6, MCP-1) and anti-inflammatory gene (adiponectin) expression; increased production of TNF-α augments this event by enhancing IL-6 and MCP-1, while 12/15-LO is involved in TNF-α associated IL-6 and MCP-1 regulation, monocyte/macrophage migration, and inflammatory factor secretion. These events play an important role in HFD induced glomerular hypertrophy and inflammatory cytokine expression.

**Table 1 nutrients-14-02743-t001:** Primer sequences used in RT-qPCR analysis and ChIP assay.

	Primers (5′→3′)	
Gene (Species)	Sense	Anti-Sense
cDNA primers		
12/15-LO (R/M)	CGCTGGCACTCTGTTTGAAGCG	TGGATGGCTATGGGCAAGA
TNF-α (M)	TGTTGCCTCCTCTTTTGCTT	TGGTCACCAAATCAGCGTTA
MCP-1 (M)	TTAAAAACCTGGATCGGAACC	GCATTAGCTTCAGATTTACGG
IL-6 (M)	ACAAAGCCAGAGTCCTTC	ACCACAGTGAGGAATGTCCAC
IL-10 (M)	CCCAAGCTTATGCCTGGCTCAGCACTGC	CGCGGATCCTTAGCTTTTCATTTTGATC
IL-1β (M)	TTGACGGACC CCAAAAG ATG	AGAAGGTGCTCATG TCCTCA
IL-12 (M)	TCTGCAGAGAAGGTCACACT	ATGAAGAAGCTGGTGCTGTA
CXCL10 (M)	CTCATCCTGCTGGGTCTGAG	CCTATGGCCCTCATTCTCAC
β-actin (R/M)	CCCTGTATGCCTCTGGTCGT	CGGACGCAGCTCAGTAACAGTCCG
ChIP primers		
IL-6 pro (M)	CGTTTATGATTCTTTCGATGCTAAACG	GTGGGCTCCAGAGCAGAATGAG
MCP-1 pro (M)	CACTAACTGAGGCCATGAACAGGTTAGTG	GCAAACCAGCACAAATGTAGCC
MCP-1 enh (M)	ATTTCCACGCTCTTATCCTACTCTG	TCACCATTGCAAAGTGAATTGG
TNF-α pro (M)	AACCGAGACAGAAGGTGCAG	TGTGCCAACAACTGCCTTTA
CypA pro (M)	GCAGGTAGGTCCTTGAGCTTGTC	CGCTAGAAGACCCTTCACCATAGCG

R, rat; M, mouse.

**Table 2 nutrients-14-02743-t002:** Physiological parameters in experimental groups.

	WT + SFD	WT + HFD	LOKO + SFD	LOKO + HFD
Fasting glucose 1 wk (mg/dL)	137 ± 22.0	128 ± 21.2	126 ± 22.1	127 ± 19.7
Fasting glucose 8 wk (mg/dL)	143 ± 26.5	189 ± 33.7	141 ± 26.3	177 ± 31.6
Fasting glucose 14 wk (mg/dL)	155 ± 29.7	243 ± 46.4 #	144 ± 34.6	226 ± 40.1 †
Serum creatinine 14 wk (mg/dL)	0.22 ± 0.02	0.31 ± 0.03	0.20 ± 0.01	0.28 ± 0.02
Body weight 1 wk (g)	20.57 ± 0.53	20.43± 1.27	23.00 ± 0.89	22.43 ± 1.62
Body weight 8 wk (g)	26.57 ± 0.98	34.57 ± 2.94 $	30.17 ± 1.17	35.86 ± 3.34 ‡
Body weight 14 wk (g)	29.57 ± 0.79	45.14 ± 2.91 $	31.83 ± 1.47	44.29 ± 4.72 £
Kidney weight 14wk (g)	0.44 ± 0.02	0.60 ± 0.02 $	0.48 ± 0.03	0.59 ± 0.03 £
K/B (%)	1.492 ± 0.049	1.340 ± 0.083 *	1.517± 0.081	1.337 ± 0.16 †
K/T (g/cm)	0.224 ± 0.010	0.316 ± 0.017 $	0.247 ± 0.012	0.310 ± 0.018 £
Urinary volume 1 wk (mL)	2.67 ± 0.67	3.10 ± 0.48	2.85 ± 0.63	3.50 ± 0.56
Proteinuria 1 wk (mg)	21.39 ± 3.88	22.53 ± 4.89	24.17 ± 3.37	22.68 ± 4.22
Urinary ACR 1 wk (μg/mg)	35.19 ± 7.01	34.40 ± 10.29	33.40 ± 11.14	36.08 ± 7.18
Urinary volume 8 wk (mL)	3.59 ± 0.53	2.53 ± 0.56 $	4.40 ± 0.58	2.49 ± 0.38 †
Proteinuria 8 wk (mg)	21.53 ± 4.10	45.53 ± 13.33 #	24.16 ± 5.24	33.41 ± 7.63 †
Urinary ACR 8 wk (μg/mg)	48.70 ± 10.38	85.66± 20.72 *	56.32 ± 15.97	59.10 ± 12.61
Urinary volume 14 wk (mL)	2.86 ± 0.66	1.36 ± 0.59 #	4.35 ± 1.31	1.47 ± 0.49 ‡
Proteinuria 14 wk (mg)	25.62 ± 6.33	42.82 ± 15.64 *	24.86 ± 5.30	33.74 ±9.54
Urinary ACR 14wk (μg/mg)	46.63 ± 12.78	106.35 ± 22.78 *	32.67 ± 7.41	72.51 ± 23.97 †

WT (C57BL/6) and LOKO mice were fed with standard fat diet (SFD) or high fat diet (HFD). Values are shown as means ± SEM; *n* = 6/group. Physiological indices were first measured at 1 week (wk) of high fat diet. * *p* < 0.05, # *p* < 0.01, $ *p* < 0.001 vs. WT + SFD; † *p* < 0.05, ‡ *p*< 0.01, £ *p* < 0.001 vs. LOKO + SFD. K/B, kidney weight/body weight ratio; K/T, kidney weight/tibial length ratio; ACR, albumin/creatinine ratio.

## Data Availability

The data presented in this study are available on request from the corresponding author.

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
