# Peer review of "Effects of Inflammatory Factor Expression Regulated by 12/15 Lipoxygenase on Obesity-Related Nephropathy"

_nutrients, 2022, doi:10.3390/nu14132743_

Round 1
Reviewer 1 Report
In this article, Liu et al elucidate the role of 12/15 LO in obesity-related nephropathy. For this, they primarily utilize 12/15 LO knockout (KO) mice fed with a high-fat diet. They observed enhanced mRNA expression of pro-inflammatory factors including TNFa, IL6, and MCP-1, with a high-fat diet; however, 12/15 LO KO led to reduced expression of these markers. Furthermore, ChIP analysis also showed that treatment of mesangial cells (MCs) with 12-HETE (a product of 12/15 LO) also enhanced H3K4me on the promoters of the proinflammatory markers. Overall, they conclude that 12/15 LO may regulate the expression of inflammatory markers in obesity-related nephropathy. Although these are some interesting findings, there are a lot of gaps, especially mechanistic insights and protein studies that need to be filled to make the study acceptable.
- The authors have quantified the mRNA expression of proinflammatory markers. Although transcriptional regulation is a known effect of 12-LO signaling, a lot of literature suggests changes at translational and post-translational levels as well. Hence, the authors need to show supporting evidence at the protein levels (by either flow cytometry, western, or ELISA) throughout the manuscript as well.
- The authors have been selective in terms of assessing proinflammatory markers (TNFa, IL6, and MCP-1). However, other pro-inflammatory markers including IL-12, IL-1b, CXCL10, CXCR3 (on innate immune cells), NF-kB, etc. have been suggested targets of 12-LO signaling as well. The authors need to check these other markers at the RNA and protein levels as well.
- In fig 1, the authors have transfected the MCs with a vector containing 12/15 LO. The experiment must be repeated at least thrice and the protein levels need to be quantified. At this stage, the protein level of 12/15 LO doesn’t seem to be much altered with transfection. Also, the protein level analysis of pro-inflammatory and anti-inflammatory markers shown in Fig 1F must be performed.
- In Fig2, the authors have studied the role of 12/15 LO in RAW cells (a macrophage cell line). Although an interesting study, the authors haven’t explained the rationale behind this experiment. It’s an interesting study because it has been shown before that 12/15-LO affects myeloid cell migration (PMID: 34128835). The authors need to study if they see alterations in macrophage/monocyte migration in their model as well. Furthermore, macrophages are also known to be a part of adipose tissue inflammation in obesity models (PMID: 30229891). So the authors need to study the abundance of macrophages in the adipose tissue or SVF of 12/15 LO KO vs WT mice fed with HFD.
- In fig 2, check protein expression of proinflammatory markers (TNFa, IL6, MCP-1, IL-12, IL-1b, CXCL10, CXCR3 (on macrophages), NF-kB, etc as well.
- In figs 3 and 4, again the authors need protein expression.
- In fig 3 and 4, the figures don’t address the core mechanism by which TNFa and 12/15 LO are linked. Also, it is unclear whether TNFa is upstream or downstream of 12/15 LO signaling. Does activation of TNFa promote 12/15 LO expression? Or is it the other way around? A proposed pathway would be useful.
- In Fig 5, the authors do not explain the rationale behind studying histone methylation. Furthermore, the mechanism is again missing. How does 12-HETE affect histone methylation? Does it act like a transcription factor or activate another factor? Experiments with appropriate controls answering these questions are warranted. Also, CHIP analysis on the promoters of the aforementioned additional proinflammatory factors needs to be performed.
- In fig 7 and 8, mechanistically how does 12/15 LO affect adiponectin, TGF-beta, and PA-1 levels?
- It’s important for the authors to discuss the potential roles of other 12 LO metabolites and their roles in obesity-induced nephropathy. There are interesting recent review articles that speak volumes about different substrates and products and their roles in inflammation (PMID: 34064822, PMID: 32540644). The authors need to cite these articles and discuss the pitfalls and considerations and their studies.
Minor:
- The authors need to work on graphical representations. They need consistency with data presentation, font styles, and font sizes. Furthermore, they need to display individual points on the graphs.
- Fig 8A, the figures are missing scale bars.
- Fig 8B, Mesangial is spelled incorrectly.
Author Response
Please see the attachment, thank you.

Reviewer 2 Report
The manuscript titled “Effects of inflammatory factor expression regulated by 12/15 lipoxygenase on obesity-related nephropathy” rises important question of the involvement of 12/15 LO in the development of obesity and obesity-dependent inflammation and insulin resistance. Few important findings from this manuscript I would like to underline: 1) the role of 12/15 LO metabolite 12(S)-HETE in the development of these pathological processes and 2) 12(S)-HETE - TNF-α axis that play a leading role in the pathology.
1) “Abstract”. A role of 12/15 LO in the development of insulin resistance and the mechanism of this involvement has to be mentioned in the abstract.
Materials and Methods
2) Page 3, line 124. The abbreviation ECM has to be explained.
3) 2.8. Histologic Staining and Morphometric Analyses page 3, line 125. Protocol for the tissue processing, cutting and staining has to be provided in details.
4) 2.4. “Glucose Tolerance Test (GTT)”. Method of blood sampling and the area of it should be described.
Results
1) Page 3. Please remove lines 135-137 from the text.
2) Page 4, 149. Use of 12/15-LO DNA vector transfection and it source have to be described in the "Methods” section
3) Page 5, line 176. “Treatment of MCs and monocyte (MO) cell line RAW264.7 …” Utilization of monocyte cell line should be described in the "Methods section". Also, authors should explain why MO have been studied as a parallel cell type. If MO were necessary for research results obtained with MO have to be more clearly discussed.
4) Figure 2, line 204, legend. It will be interesting to evaluate and describe if siRNA without PA reduce baseline expression of cytokines MCs treated with si-NTC.
Legend to Figure 3, line 232. The abbreviation SD has to be explained. Also for the figure 3B the comparisons between WT and LOKO should be made also for the demonstration that KO induces significant effect. As well, check IL-6 in fig 3B. Is this really no statistically significant difference between LOKO and LOKO+TNF?
5) “3.4. Effect of TNF-α Inhibition on 12/15-LO-regulated Inflammatory Factors”. Why effects of si-TNF were studied on MO but not on MG, despite the fact that TNF affects MG? If authors really work on MO not MG, they have to be more clear in the “Discussion”.
Discussion
1) Please remove lines 477-480.
Author Response
Please see the attachment, thank you.

Round 2
Reviewer 1 Report
The authors have addressed many if not all of my concerns. Adding protein data certainly enhances the quality of the work. For final submission, the minor changes authors should make include :
a. Display all the data points on the graphs in all figures
b. Include a few statements on the caveats of their study
c. Do thorough proofreading of the article by a professional to avoid grammatical errors.